# Steady-State Levels of Cytokinins and Their Derivatives May Serve as a Unique Classifier of Arabidopsis Ecotypes

**DOI:** 10.3390/plants9010116

**Published:** 2020-01-17

**Authors:** Zuzana Samsonová, Nagavalli S. Kiran, Ondřej Novák, Ioannis Spyroglou, Jan Skalák, Jan Hejátko, Vít Gloser

**Affiliations:** 1Department of Experimental Biology, Faculty of Science, Masaryk University Kamenice 5, Brno 62500, Czech Republic; zuzka.samsonova@gmail.com (Z.S.); naasuki@gmail.com (N.S.K.); 2Faculty of Science, Palacký University and Laboratory of Growth Regulators, Institute of Experimental Botany, The Czech Academy of Sciences, Šlechtitelů 27, Olomouc 78371, Czech Republic; ondrej.novak@upol.cz; 3Plant Sciences Core Facility, CEITEC MU, Central European Institute of Technology, Masaryk University, Kamenice 5, Brno 62500, Czech Republic; ioannis.spyroglou@ceitec.muni.cz; 4Functional Genomics & Proteomics of Plants, CEITEC MU, Central European Institute of Technology, Kamenice 5, Brno 62500, Czech Republic; jan.skalak@ceitec.muni.cz (J.S.); hejatko@sci.muni.cz (J.H.)

**Keywords:** abscisic acid, cytokinins, cytokinin glucosides, cytokinin metabolism, indole-3-acetic acid, single nucleotide polymorphism

## Abstract

We determined steady-state (basal) endogenous levels of three plant hormones (abscisic acid, cytokinins and indole-3-acetic acid) in a collection of thirty different ecotypes of *Arabidopsis* that represent a broad genetic variability within this species. Hormone contents were analysed separately in plant shoots and roots after 21 days of cultivation on agar plates in a climate-controlled chamber. Using advanced statistical and machine learning methods, we tested if basal hormonal levels can be considered a unique ecotype-specific classifier. We also explored possible relationships between hormone levels and the prevalent environmental conditions in the site of origin for each ecotype. We found significant variations in basal hormonal levels and their ratios in both root and shoot among the ecotypes. We showed the prominent position of cytokinins (CK) among the other hormones. We found the content of CK and CK metabolites to be a reliable ecotype-specific identifier. Correlation with the mean temperature at the site of origin and the large variation in basal hormonal levels suggest that the high variability may potentially be in response to environmental factors. This study provides a starting point for ecotype-specific genetic maps of the CK metabolic and signalling network to explore its contribution to the adaptation of plants to local environmental conditions.

## 1. Introduction

Plant evolution has been shaped by many factors including environmental conditions or the size and geographical distribution of populations of individual species. It has been shown that large variability in morphology and functions may exist even among populations of the same species. An increased incidence of single nucleotide polymorphisms (SNPs) has been reported among genes mediating interaction with the external environment [1]. It turns out that almost 10% of genes coding proteins contain a significant change such as an altered reading frame or a premature stop codon [1]. Nevertheless, the extent to which adaptation to a specific region is reflected in the phenotype of particular plant ecotypes is still poorly understood and the precise effect of the genomic changes on the individual phenotype is not clear. But it is widely presumed that some of these changes contribute to increased phenotypic variability.

As with many other traits, basal hormone levels in the plant are genetically determined. Given their key regulatory function, even small, genetically determined differences can play a significant role in the control of intraspecific variability of plant development or adaptive responses. The variability of basal hormone content due to genetic divergence is not known. Further, it is unclear whether there is a relationship between increased incidence of SNPs in genes related to interactions with the environment and the particular climatic and environmental conditions prevailing in the geographical place of origin of the particular intraspecific population. In other words, it is still not clear if the differences in phenotype observed in various subpopulations of a given species (varieties, ecotypes) may be a direct result of the selective pressure on the developmental and adaptive determinants of that population.

Natural variation among ecotypes of *Arabidopsis thaliana* is increasingly drawing the attention of researchers [2,3,4,5,6]. *A. thaliana* is an important model plant and the knowledge of variations among ecotypes and the derived mutant plants is valuable for many researchers. Nevertheless, surprisingly little is known about differences in physiological functions and plant-environment interactions among the ecotypes and how these differences may be affected by variations in hormone levels.

Adaptation of plants to different environmental conditions is tightly connected to plant hormones as they are the principal mediators of plant developmental responses. As with other traits, differences exist among species in hormone levels and, consequently, in their response to environmental conditions [7]. It has been shown that significant differences exist in this respect not only among different species but also among individual ecotypes within the same species such as *A. thaliana.* Our preliminary experiments indicated that the differences exist in levels of cytokinins (CKs) and their metabolites among ecotypes even in plants grown under stable and identical environmental conditions [8]. Although hormone levels change in response to variations in the environment, the dissimilarities in inherent basal concentrations may determine potential differences in the dynamics or even in the magnitude of the response to such environmental factors. Examples of plant hormone-mediated responses to the environment without any change in hormonal level, but due to factors modulating hormonal signal such as pH, have already been well documented for abscisic acid (ABA) [9].

In our work, we used a collection of thirty different *A. thaliana* ecotypes (Table 1) that represent a wide genetic variability within the species based on SNPs [1]. The group also included other ecotypes frequently used in experimental work (e.g., Ws-0, Col-0, Ler-1). We determined steady-state (basal) levels of three major plant hormones-cytokinins, ABA and auxin (indole-3-acetic acid—IAA) in this set of ecotypes grown under optimal and stable environmental conditions. We hypothesised that although a large variability exists among ecotypes in the levels of all major plant hormones even when all plants are cultivated under the same environmental conditions, significant patterns of hormone content exist within the whole group of ecotypes. Using advanced statistical methods, we analysed the extent to which hormone levels can specifically identify a particular ecotype. We also explored the possible relationships of hormone levels with the environmental conditions of the geographical site where the respective ecotype was originally collected.

## 2. Results and Discussion

### 2.1. Inherent Variation in Hormone Content

Similarly to other compounds, also hormonal content in the tissue can be influenced by the relative contribution of various cell types. Therefore, it is also influenced by the morphology and developmental stage of the plant body and the inherent differences in the rates of growth and development should be considered when comparing the set of ecotypes. Our preliminary experiments, however, showed that in the early stage of growth the differences in size and morphology among ecotypes are small. Only a few ecotypes were significantly different in the number of leaves (Br-0, Cvi-0, RRS-10, Tsu-1) than the rest of the tested group (Appendix A). Similarly, only a few ecotypes significantly differed in root length (Appendix A). Additionally, we found no statistically significant correlation between hormonal content in selected ecotypes and either ecotype morphology or inherent developmental rate of the ecotype.

Given that the plants were grown under identical and stable cultivation conditions resulting in uniform growth rate and development/morphology of ecotype majority, the variability in hormone contents that we observed among the ecotypes was surprisingly high. The content of CKs varied between 150 to 500 pmol g^−1^ in both shoot and root (Figure 1). Interestingly, the Ws-0 ecotype that has frequently been used in genomic analyses and other experimental work had the smallest content of CKs in both root and shoot of all the examined ecotypes and appeared to be an extreme case in this respect. Variation in CK content was comparable in shoot and root and ecotypes with high CK content usually showed similar levels in both organs (e.g., Hr-5, Ct-1). It was previously shown that high CK content, particularly in the shoot, may be beneficial for growth and drought stress tolerance [10]. In this context, by far the most frequently used ecotype—Col-0 had the highest proportion of basal CK concentration in the shoot of all the ecotypes examined (Appendix A). Seventy per cent of examined ecotypes had a higher CK content in roots, which fits well with previous observations suggesting a dominant role for roots in CK biosynthesis [11,12,13]. Generally, we found only a rather small variation of the ratio of CKs contents in the root and shoot in our set of ecotypes (Appendix A). A possible explanation for this could be similar transport rates of root-derived CKs to the shoot in the majority of ecotypes. Interestingly, Col-0 revealed the lowest root-to-shoot ratio, being a consequence of both low root and high shoot CK content measured in Col-0 seedlings.

Compared to CKs, the content of IAA differed considerably between the shoot and root (Figure 2, Appendix A). In shoots, the IAA content varied between 80 and 250 pmol g^−1^ whereas in roots the amount of IAA was much greater, typically between 220 to 1000 pmol g^−1^, also with greater variation (approx. 4.5 fold) than in the shoot. Interactions of IAA and CKs are currently a target of intensive research [14]. It has long been known that the ratio of auxin to cytokinin determines the differentiation of plant tissues [15,16]. We tested the possible importance of the relationship between IAA and CK contents among the ecotypes. Although the ratio varied more than seven-fold (Appendix A) there was no significant correlation either in shoots (r = −0.288; *p* = 0.1228) or in roots (r = −0.2374; *p* = 0.2065). In general, some ecotypes showed systematically higher levels (in the range of upper 50% among ecotypes examined) of all analysed hormones in both shoots and roots (Mt-0, Wei-0, Ta-0, RRS-10, Sha), whereas others showed systematically lower levels (lower 50% among ecotypes examined-Bor-4, Col-0, Ts-1, Ler-1). The highest levels of IAA typically associated with the highest IAA/CK ratios (Figure 2, Appendix A).

The content of ABA in both shoot and root was low, typically ranging between 5 and 30 pmol g^−1^ with an approximately ten-fold variation among ecotypes (Figure 3). We observed greater variation in the shoot. Generally, low ABA contents are probably related to high water availability and high humidity inside cultivation dishes [17]. In many ecotypes, the slightly higher ABA levels in the root were also accompanied by higher ABA content in the shoot. About two-thirds of the ecotypes showed higher ABA content in roots compared to shoots (Appendix A), which fits with the view that roots are the dominant source of ABA as well. Moreover, in five of the ecotypes the ABA content in the root was more than double that in the shoot (Got-7, Tsu, NFA8, Ta-0, Sha; Appendix A). Elevated ABA levels only in the root (but not in the shoot) were previously shown to be beneficial for plant productivity since both the hydraulic conductance of roots and stomatal conductance are high, and that favours plant gas exchange [18]. Elevated ABA in the root may also support plant drought tolerance, as suggested by Ghanem et al. [19].

In ecotypes with high levels of IAA, we frequently observed lower levels of ABA (Figure 2 and Figure 3) most likely as a result of the mutually negative interactions between metabolic pathways of the two hormones [20]; however, this trend was not significant. Therefore, we also examined the variation in the ratio of IAA and ABA among the ecotypes (Figure 4). The interplay between IAA and ABA can affect the development of the root system, thereby influencing plant (drought) stress tolerance [19]. The ratio reflected considerably higher content of IAA, and it varied about fivefold. In some ecotypes (Ta-0, Cvi-0), however, the ratio was considerably shifted in favour of ABA content (Figure 4).

### 2.2. Hormone Levels as an Ecotype-Specific Trait

Using several statistical approaches we explored the significance of basal hormonal levels and their ratios in roots and shoots as a specific identifier of the ecotype. The Random Forest classifier with the significance measure of Mean Decrease in Impurity (MDI) [21] showed that at the level of total hormone content, CKs and less so ABA (specifically for the Ta-0 ecotype) can be significant identifiers for a small number of ecotypes (see Appendix B, Table A1 and Table A2). In the next step of our analysis, we used the complete data set including all metabolites in the CKs group. This approach produced much more precise and significant results.

Classification analysis revealed that glycosylation contributes significantly to the setting of ecotype-specific CK levels in both roots and shoots during early seedling development. CK glucosides have the highest accuracy even when a similar number of trait variables are considered for classification (see Table 2). The classification of ecotypes was significantly improved for both shoots and roots when glucosides were included in the analysis and improved classification can also be inferred from the fact that CK glucosides seem to play the most important role in the final Random Forest classifier (Table 2). The total classification accuracy of the classifier (using repeated hold-out cross-validation) was 72.1% in roots and 72.9% in shoots (Table 2). Furthermore, the importance of individual metabolites for classification can be seen from the MDI index (Table 3). The finding that iPR is an important classifier in both shoots and roots (Table 3) provides support to the hypothesis that this compound is the principal form in which root-synthesized CKs are delivered to the shoot [22,23,24,25,26,27].

It should be mentioned that Random Forest classifier does not require any prior data scaling (normalization, standardization) as it consists of decision trees, which in turn consist of splitting rules starting at the top of the tree. Therefore, since the magnitudes of the features are irrelevant in methods which use splitting rules, there is no bias originating from the difference in the feature magnitudes (absolute values) and consequently no need for scaling as it would give the same results [28]. Thus, the fact that CK glucosides are the most abundant form of endogenous CK, usually being an order of magnitude higher than the other CK forms, does not bias the determination of their importance compared to other, less abundant CK metabolites.

Since ecotypes in our study represent large genetic diversity in *A. thaliana*, our finding that levels of CK glucosides represent a powerful classification trait can be taken as generally valid for the whole species. We see the same trend even among those ecotypes that are classified as almost perfect outliers (Bay-0, Cvi-0, Kon, Ler, RRS-7, Van-0, Ws-0, Ws-2; see Appendix B for more information). This demonstrates the significance of glucosides as an important contributor to ecotype-specific variability of endogenous CK levels. It is apparent from other experiments that for individual physiological responses it is important to consider not just the levels of active CKs, but also those of metabolites that do not directly trigger CK receptor action [29]. From our survey, we see that in the normal growth of the plant (at least during early seedling development) the entire CK metabolic network, including inactivated metabolites, plays a significant role in determining CK levels, and by extension, CK response. CK glycosylation was previously demonstrated as one of the first and highly potent mechanisms allowing the removal of large amounts of endogenous CKs from biologically active CK pool [30,31,32,33,34]. Thus, CK modification by glycosyltransferases might represent a crucial mechanism controlling the metabolism and function of these compounds as along with CK transport and the specific distribution of CKs in cells and tissues which further affect the development of plants [35].

It is well known that even minor local and transient changes in CK levels can have profound physiological effects [36,37,38,39]. Thus it is highly likely that the specificity of CK action depends to a significant extent on the determination setting of CK levels with contributions from the entire network of CK conversions. Therefore, the diverse mechanisms that affect this determination must be considered to get a full picture of the CK role in any physiological response. Such mechanisms include not only the conversions of CK from one form to another but also other mechanisms underlying the availability of individual CK substrate forms and their cognate enzymes. Furthermore, the CK homeostatic mechanisms are very dynamic and include processes such as cell-to-cell transport and sub-cellular sequestration or compartmentation of CK metabolites as well as genetic variability of TCS components which in turn further modulate the primary CK response.

Our study lays the groundwork for a more detailed exploration of the contributors to the ecotype-specific levels of CKs observed in our study. Our data indicate that the activities of enzymes that are responsible for CK metabolic conversions, particularly those responsible for glycosylation/deglycosylation, vary in an ecotype-specific manner, and this variation gives rise to the observed importance of glucosides as the strongest classifier. Further, it is highly likely that ecotype-specific variations exist in the sensitivity to CKs, and a thorough explication of this variability could form part of a future analytical study.

### 2.3. Hormone Levels and the Site of Origin of Ecotype

Besides the inherent hormonal settings among ecotypes, we also explored the relationship between basal hormonal levels and the geographical location and environmental conditions prevalent in the sites of origin of the respective ecotypes.

Plant response to stress is frequently associated with an increase in the content of ABA and also the ratio between ABA and CKs that combine to regulate stomatal aperture size [40,41]. This shift in hormonal content contributes to the balance between plant growth rate and availability of resources (e.g., water, nutrients), particularly under stress-inducing conditions [42]. We indeed found a large variation in the CK/ABA ratio among ecotypes in both shoots and roots (Figure 5). Detailed analysis of data discovered more significant relationships at the organ level. There was a positive correlation of mean ABA content in the shoot with the mean temperature at the site of origin (r = 0.404, *p* = 0.02688; Figure 6A). The correlation between the ABA/CK ratio in the shoot and the site temperature was also significant (r = 0.4317, *p* = 0.0172, Figure 6B). Moreover, there was also a significant positive correlation between the ABA shoot content and altitude above sea-level of the site of origin (r = 0.4, *p* = 0.031); however, this relationship was most likely mediated by the strong correlation between the altitude and the mean site temperature. Effects of the temperature at the site of origin on basal hormonal balance can be further highlighted by classifying the sites based on mean temperature (Figure 7). We labelled the sites as WARM (T > 15.5 °C) and COLD (T < 15.5 °C). Results of a t-test and a Welch test revealed that places with a mean temperature higher than 15.5 °C tend to have higher mean ABA values for the shoots (*p* = 0.08 for both tests). No other correlations of basal hormone levels with geographical and environmental factors showed any significant relationships (Appendix A).

## 3. Materials and Methods

### 3.1. Plant Material

A collection of 30 selected ecotypes of *Arabidopsis thaliana* based on observations by Clark et al. (2007) was used in this study (Table 1) [1]. Tamm-2 from the original Clark collection was not included due to unsuccessful seed propagation. Seeds were obtained from NASC (Nottingham Arabidopsis Stock Centre, Nottingham, UK) and propagated in stable greenhouse conditions with supplemented light.

### 3.2. Plant Cultivation and Sampling

Before sowing, all seeds were sterilized for 5 min in a solution of 75% EtOH with 0.1% Triton-X-100 and dried thoroughly. Seeds of each ecotype were sown separately in square Petri plates (12 × 12 cm), 15 seeds per plate and 20 plates per ecotype.

The cultivation medium was half-strength Murashige and Skoog (MS) without sucrose that contained: MS macro and microelements (Duchefa, Haarlem, The Netherlands) with MES buffer (500 mg L^−1^) in redistilled water; pH was corrected by 1 M KOH (5.7–5.8). The medium was solidified using 1.2% agar.

A sterile fabric with ultrafine mesh (Uhelon, Silk & Progress, Brněnec, CZ) was placed on the plated, solidified medium a few hours before sowing to facilitate harvesting. After sowing, the plates were sealed with vapour permeable tape and stored at 4 °C in the dark for three days for stratification. On the fourth day, plates were placed vertically in a cultivation box with stable environmental conditions: photosynthetic photon flux density 120 µmol m^−2^ s^−1^ supplied with fluorescent lamps (Osram Fluora TLD 36W, Osram AG, Munich, Germany) with photoperiod 16 h, temperature 21/19 °C day/night and air humidity 40 and 50% during day and night respectively.

Plants were harvested on the 21st day after sowing (DAS). The timing of harvest was based on our preliminary experiments showing only minor differences among ecotypes in morphology 14 to 21 DAS (Appendix A) and, simultaneously, providing a sufficient amount of biomass for analyses. Plants typically had 6 to 8 leaves and root systems with only a small number of short lateral roots. Roots and shoots were weighed separately and immediately flash-frozen in liquid nitrogen and stored at −80 °C until hormone quantification. Means of 3–4 replicate samples per organ and ecotype (biological replicates) were analysed. Each sample contained several shoots (3–8) or roots (20–30).

### 3.3. Quantification of Hormones

Small amounts of tissue (20 mg) were homogenized and extracted in 1 mL of modified Bieleski buffer (60% MeOH, 10% HCOOH and 30% H_2_O) together with a cocktail of stable isotope-labeled internal standards (0.25 pmol of CK bases, ribosides, *N*-glucosides, 0.5 pmol of CK *O*-glucosides and nucleotides, 5 pmol of ^2^H_5_-IAA and ^2^H_5_-ABA per sample). The extracts were purified using two solid-phase extraction columns, an octadecylsilica-based column (C18, 500 mg of sorbent, Applied Separations) and after that an Oasis MCX column (30 mg/1 mL, Waters, Milford, MA, USA) [43]. Analytes were eluted by a three-step elution using 60% (*v*/*v*) MeOH, 0.35 M NH_4_OH aqueous solution and 0.35 M NH_4_OH in 60% (*v*/*v*) MeOH. Samples were subsequently analysed using an ultra-high performance liquid chromatography (UHPLC) system coupled to a tandem quadrupole mass spectrometer (MS/MS) equipped with an electrospray interface (ESI) [44,45]. Quantification was performed by Masslynx software (Waters) using a standard isotope dilution method with stable isotope-labelled internal standards as a reference.

The following hormones and their derivatives were quantified: IAA, ABA, CKs. IAA was chosen as it is the predominant natural form of auxin. CKs were presented as a sum of all major and minor components detected in each of their metabolic groups (bases, ribosides, nucleotides, *O*-glucosides, *N*-glucosides. Namely: *trans*-zeatin (*t*Z), *trans*-zeatin riboside (*t*ZR), *trans*-zeatin-*O*-glucoside (*t*ZOG), *trans*-zeatin-*o*-glucoside riboside (*t*ZROG), *trans*-zeatin *N*7-glucoside (*t*Z7G), *trans*-zeatin *N*9-glucoside (*t*Z9G), *trans*-zeatin riboside-5′-monophosphate (*t*ZRMP), *cis*-zeatin (*c*Z), *cis*-zeatin riboside (*c*ZR), *cis*-zeatin-*O*-glucoside (*c*ZOG), *cis*-zeatin-*O*-glucoside riboside (*c*ZROG), *cis*-zeatin *N*9-glucoside (*c*Z9G), *cis*-zeatin riboside-5′-monophosphate (*c*ZRMP), dihydrozeatin (DHZ), dihydrozeatin riboside (DHZR), dihydrozeatin-*O*-glucoside (DHZOG), dihydrozeatin-*O*-glucoside riboside (DHZROG), dihydrozeatin *N*7-glucoside (DHZ7G), dihydrozeatin *N*9-glucoside (DHZ9G), dihydrozeatin riboside-5′-monophosphate (DHZRMP), isopentenyladenine (iP), isopentenyladenosine (iPR), isopentenyladenine *N*7-glucoside (iP7G), isopentenyladenine *N*9-glucoside (iP9G), isopentenyladenine riboside-5′-monophosphate (iPRMP).

### 3.4. Statistical Analysis and Ecotype Classification Using Random Forest

Possible relationships among various hormones and between hormone levels and environmental factors were tested by correlation analysis using the Pearson correlation coefficient. Environmental factors involved in the analysis were: Average precipitation (spring/autumn), average temperature (spring/autumn), mean day-length, geographical location on the globe (latitude and longitude) and elevation above sea level. The data used were obtained for sites in the closest proximity to the site of origin of respective ecotype where the data were freely available. Significance of differences among sites with different temperatures was also tested by *t*-test. Despite the fact that the samples are extremely small, the effect sizes are large and thus, we examined differences using both *t*-test and Welch test (unequal variance *t*-test) [46,47]. The classification analysis was based on the Random Forest classifier method. This method is particularly useful when there are more than two categories in a response variable—e.g., ecotypes [21,48]. A more detailed description of the methodology of classification analysis is provided in the Appendix A. All statistical tests were performed using the R software [49].

## 4. Conclusions

We found significant variations in the basal hormone levels among a wide range of *A. thaliana* ecotypes. Considering the previously demonstrated role of hormonal regulation in stress response [7], the large variability in basal hormone levels, as well as some significant relationships with temperature as a key environmental factor, might imply inherent differences among ecotypes in drought stress tolerance [50]. We also clearly showed that CK levels-uniquely among the hormones tested can be used as an ecotype classifier. These ecotype-specific levels of CKs might further affect the two-component system (TCS) to regulate the primary response of *A. thaliana* to internal as well as to environmental stimuli. Nevertheless, the actual CK pool does not reflect the activity of TCS, which carries SNPs among the contrasting ecotypes to fine-tune CK signalling, suggesting avenues for further research of the natural genetic variability of TCS components. Natural polymorphism in TCS components might substantially affect the final CK response as shown for instance in mutagenic studies on CK sensor histidine kinases, which identified mutations leading to either impaired kinase activity or conversely to CK-independent activation [51,52,53,54].

The role of CK-glucosides in the regulation of plant growth and development is still unclear. Our data, however, suggest that the content of CK-glucosides might be a trait specific to individual ecotypes. The pool of CK-glucosides that can potentially contribute to levels of active CK appears to be larger than previously expected. The generally accepted hypothesis that *N*-glucosylation irreversibly inactivates CKs was recently put into question [55]. Thus, release of active CKs from CK glucosides including the *N*-glucosides, may substantially affect the whole metabolic pattern. Our study highlights the necessity of investigations on the functional importance and underlying molecular mechanisms of genetic variability of CK metabolism, particularly CK glucosylation as well as conversions of CK glucosides to free CKs.

## Figures and Tables

**Figure 1 plants-09-00116-f001:**
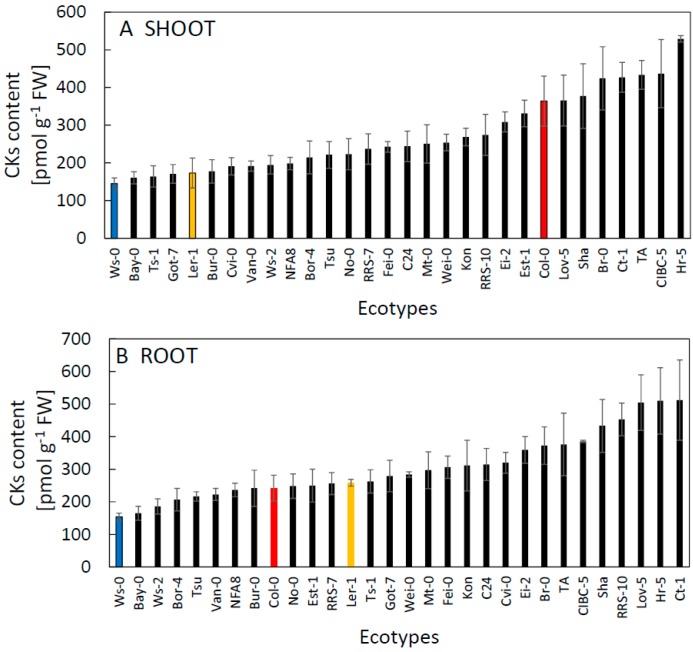
The content of CKs in shoot (**A**) and root (**B**) of thirty *A. thaliana* ecotypes grown 21d in controlled conditions. Commonly used ecotypes are marked in colour (Red = Col-0, Blue = Ws-0, Orange = Ler-1). See Table 1 for the full list of ecotypes. Means ± SD.

**Figure 2 plants-09-00116-f002:**
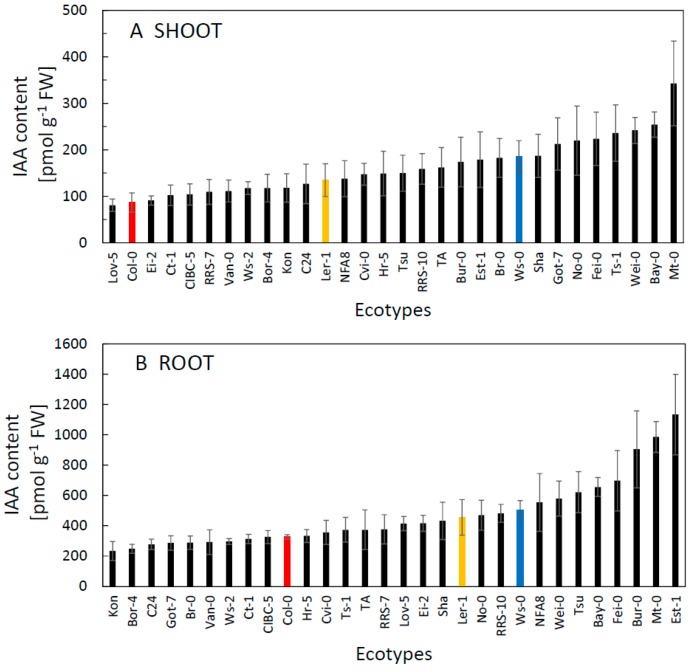
The content of IAA in the shoot (**A**) and root (**B**) of thirty *A. thaliana* ecotypes grown 21d in controlled conditions. Commonly used ecotypes are marked in colour (Red = Col-0, Blue = Ws-0, Orange = Ler-1). See Table 1 for the full list of ecotypes. Means ± SD.

**Figure 3 plants-09-00116-f003:**
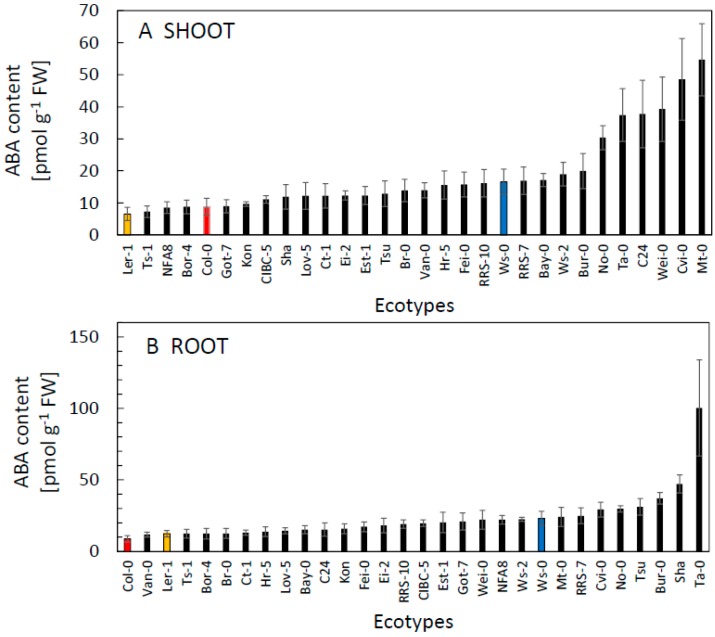
The content of ABA in the shoot (**A**) and root (**B**) of thirty *A. thaliana* ecotypes grown 21d in controlled conditions. Commonly used ecotypes are marked in colour (Red = Col-0, Blue = Ws-0, Orange = Ler-1). See Table 1 for the full list of ecotypes. Means ± SD.

**Figure 4 plants-09-00116-f004:**
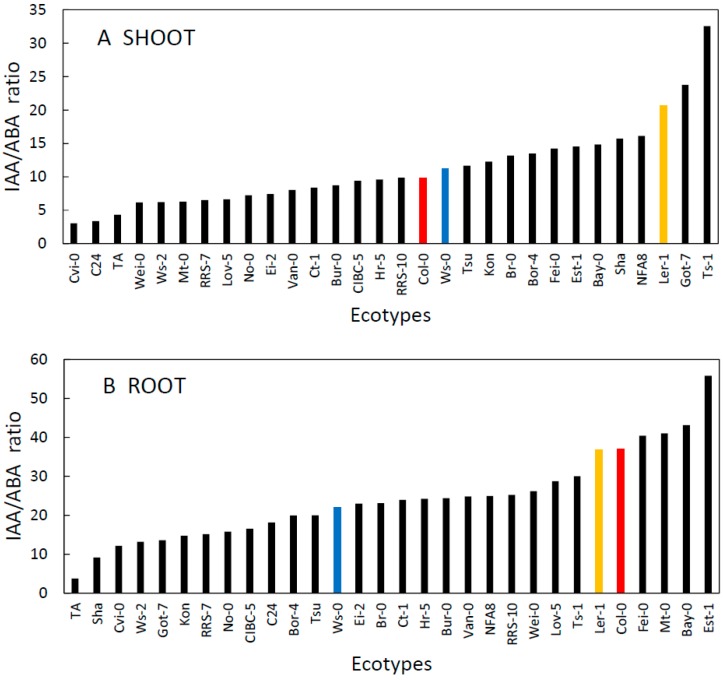
The ratio of IAA to ABA content in the shoot (**A**) and root (**B**) ranked from the lowest to the highest among thirty *A. thaliana* ecotypes grown 21d in controlled conditions. Commonly used ecotypes are marked in colour (Red = Col-0, Blue = Ws-0, Orange = Ler-1). See Table 1 for the full list of ecotypes.

**Figure 5 plants-09-00116-f005:**
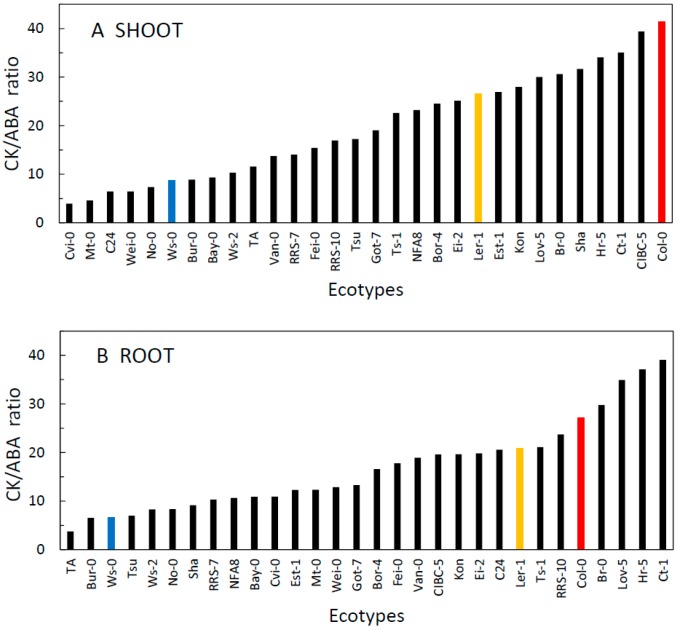
The ratio of total CKs to ABA in the shoot (**A**) and root (**B**) ranked from the lowest to the highest among thirty *A. thaliana* ecotypes grown 21d in controlled conditions. Commonly used ecotypes are marked in colour (Red = Col-0, Blue = Ws-0, Orange = Ler-1). See Table 1 for full list of ecotypes.

**Figure 6 plants-09-00116-f006:**
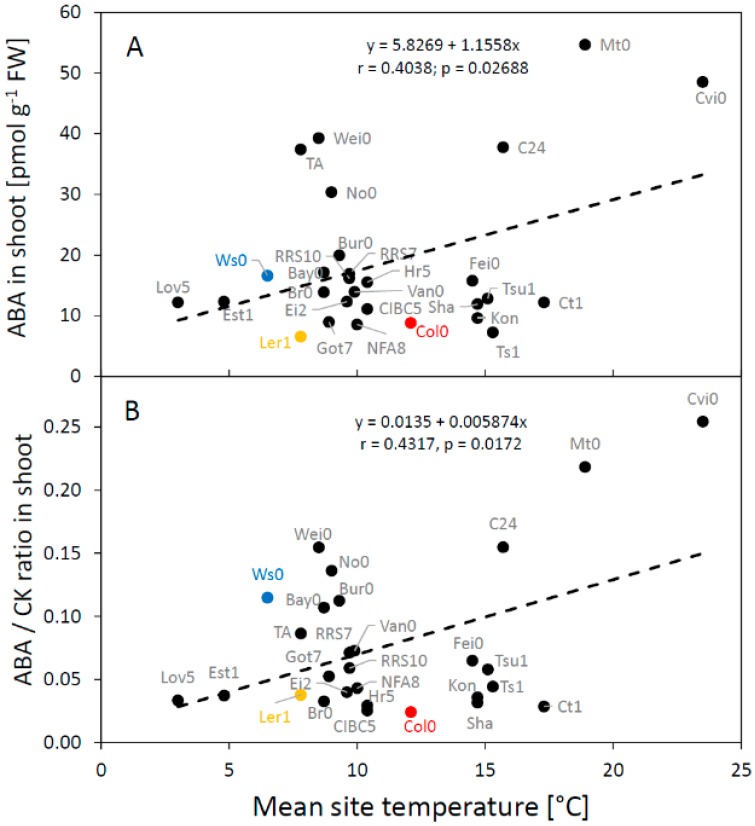
Relationships between the mean temperature at the site of ecotype origin and basal level of abscisic acid (ABA) content in the shoot (**A**) and the ratio between ABA and total cytokinin content in the shoot (ABA/CK ratio) in the shoot (**B**) in *A. thaliana* ecotypes. Commonly use ecotypes are marked in colour (Red = Col-0, Blue = Ws-0, Orange = Ler-1). See Table 1 for the full list of ecotypes.

**Figure 7 plants-09-00116-f007:**
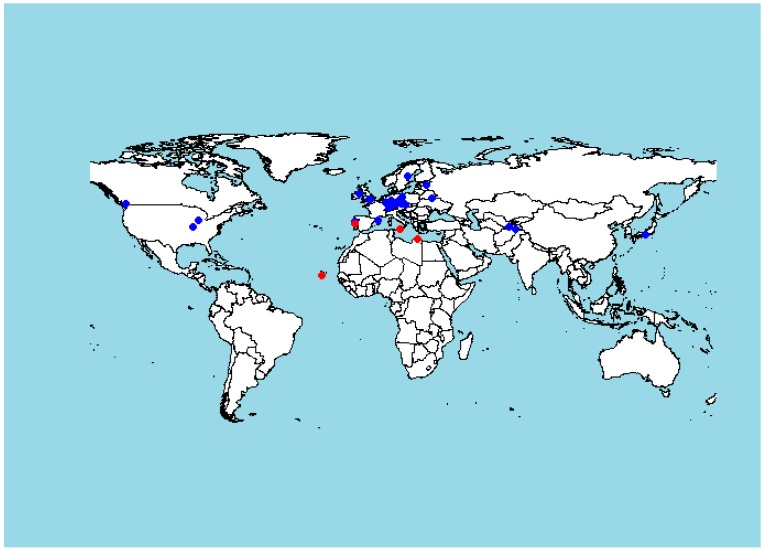
World map showing locations of sites of origin for the examined *A. thaliana* ecotypes. Colors indicate classification of sites based on the mean temperature: Red: WARM (T > 15.5 °C), Blue: COLD (T < 15.5 °C). The map was generated in R package (v. 3.3.0) using the package “Draw Geographical Maps” (2018). (https://cran.r-project.org/web/packages/maps/).

**Table 1 plants-09-00116-t001:** List of ecotypes used in this study including abbreviations, country and region of origin of each ecotype. Listed are also main environmental parameters for each site of origin: Mean year temperature in °C (Temp.), Mean total annual precipitation in mm (Prec.), Mean altitude (Altit.) and precise geographical location—Latitude and Longitude (Lat., Long.).

Ecotype	Country	Region	Temp.	Prec.	Altit.	Lat.	Long.
Bay-0	Germany	Bayreuth	8.7	490	350	49	11
Bor-4	Czech Rep.	Borky (Brno)	8.7	490	250	49.4	16.2
Br-0	Czech Rep	Brunn (Brno)	8.7	490	250	49.2	16.6
Bur-0	Ireland	Burren	9.3	805	50	54.1	−6.2
C24	Portugal	Coimbra	15.7	1014	150	40.2	−8.4
CIBC-5	UK	Ascot, Berks	10.4	594	50	51.4	−0.6
Col-0	USA	Columbia	12.1	991	50	38.3	−92.3
Ct-1	Italy	Catania	17.3	547	50	37.3	15
Cvi-0	Cape Verde Islands	Santa Cruz	23.5	70	1150	15.1	−23.6
Ei-2	Germany	Eifel	9.6	933	450	50.3	6.3
Est-1	Estonia	Tartu	4.8	589	150	58.3	25.3
Fei-0	Portugal	Santa Maria da Feira	14.5	1267	50	40.9	−8.5
Got-7	Germany	Göttingen	8.9	655	150	51	10
Hr-5	UK	Ascot, Berks	10.4	594	50	51.4	−0.6
Kon	Tajikistan	Khurmatov	14.7	568	800	38.5	68.5
Ler-1	Poland/Germany	Gorzow Wielkopolski/Landsberg	7.8	805	450	48	10.9
Lov-5	Sweden	Lövvik/Sandöverken, Harnosand	3.0	570	50	62.8	18.1
Mt-0	Libya	Martuba/Cyrenaika	18.9	96.5	150	32.3	22.5
NFA-8	UK	Ascot	10.0	570	50	51.4	-0.6
No-0	Germany	Nossen/Halle	9.0	595	150	51.1	13.3
RRS-10	USA	North Liberty, Indiana	9.7	994	250	41.6	−86.4
RRS-7	USA	North Liberty, Indiana	9.7	994	250	41.6	−86.4
Shahdara	Tajikistan	Shakdara river (Pamiro Alaya)	14.7	568	3350	37.4	71.6
Ta-0	Czech Rep.	Tabor	7.8	526	450	49.5	14.5
Ts-1	Spain	Tossa del Mar	15.3	658	50	41.7	2.9
Tsu-1	Japan	Tsukuba/Tsushima	15.1	1535	50	34.4	136.3
Van-0	Canada	Vancouver	9.9	1167	50	49.3	−123
Wei-0	Switzerland	Weiningen	8.5	1101	550	47.3	8.3
Ws-0	Belarus	Vasil’yevka	6.5	588	150	52.3	30
Ws-2	Belarus	Vasil’yevka	6.5	588	150	52.3	30

**Table 2 plants-09-00116-t002:** Overall classification accuracy of the ecotypes using Random Forest classifier with different traits using hold-out cross validation. Number of compounds used for analysis is stated in brackets.

Trait Variables Used in the Random Forest as Predictors	Accuracy (%)	
	Roots	Shoots
ABA, total CKs, IAA (3)	28.5%	32.1%
CK non-glucosides (10)	38.4%	41.9%
CK glucosides (12)	68.8%	66.5%
All CKs (22)	72.1%	72.9%

**Table 3 plants-09-00116-t003:** The importance of various cytokinin derivatives in root and shoot as trait variables for classification based on the Mean Decrease in Impurity value (MDI). The most important compounds are on the top of the list. Abbreviations: *trans*-zeatin (*t*Z), *trans*-zeatin riboside (*t*ZR), *trans*-zeatin-*O*-glucoside (*t*ZOG), *trans*-zeatin-*O*-glucoside riboside (*t*ZROG), *trans*-zeatin *N*7-glucoside (*t*Z7G), *trans*-zeatin *N*9-glucoside (*t*Z9G), *trans*-zeatin riboside-5′-monophosphate (*t*ZRMP), *cis*-zeatin (*c*Z), *cis*-zeatin riboside (*c*ZR), *cis*-zeatin-*O*-glucoside (*c*ZOG), *cis*-zeatin-*O*-glucoside riboside (*c*ZROG), *cis*-zeatin *N*9-glucoside (*c*Z9G), *cis*-zeatin riboside-5′-monophosphate (*c*ZRMP), dihydrozeatin (DHZ), dihydrozeatin riboside (DHZR), dihydrozeatin-*O*-glucoside (DHZOG), dihydrozeatin *N*7-glucoside (DHZ7G), dihydrozeatin *N*9-glucoside (DHZ9G), isopentenyladenosine (iPR), isopentenyladenine *N*7-glucoside (iP7G), isopentenyladenine *N*9-glucoside (iP9G), isopentenyladenine riboside-5′-monophosphate (iPRMP).

ROOT	MDI	SHOOT	MDI
**cZROG**	3.98783	cZROG	5.28804
**iP7G**	3.93474	iPRMP	4.60337
**iPR**	3.71329	iP7G	4.45139
**tZROG**	3.5989	iPR	4.37966
**DHZ7G**	3.58955	tZROG	3.99245
**cZR**	3.39747	DHZOG	3.87761
**tZRMP**	3.35821	cZ9G	3.80151
**cZOG**	3.305	DHZ9G	3.62936
**DHZOG**	3.2893	iP9G	3.48557
**tZOG**	3.00218	DHZ7G	3.33904
**DHZ**	3.0011	tZRMP	3.3029
**tZR**	2.9322	tZR	3.20053
**cZ9G**	2.93091	iP	3.16696
**tZ7G**	2.8261	tZ7G	3.01294
**tZ9G**	2.72127	tZ9G	2.97171
**cZRMP**	2,45831	DHZ	2.95543
**DHZ9G**	2.399	cZR	2.86284
**iPRMP**	2.39615	tZOG	2.80756
**cZ**	2.22799	DHZR	2.62581
**DHZR**	2.12067	tZ	2.46915
**tZ**	2.08467	cZRMP	2.44495
**iP9G**	2.039	cZOG	2.27142

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
