# Peer review of "Steady-State Levels of Cytokinins and Their Derivatives May Serve as a Unique Classifier of Arabidopsis Ecotypes"

_plants, 2020, doi:10.3390/plants9010116_

Round 1

Reviewer 1 Report

This is potentially very interesting paper what try to linked ecotypes with hormone contents.

Authors used very interesting approach to classify Arabidopsis ecotype according to hormone contents and find that cytokinin may serve as a best marker for this classification..

However, there are some methodological mistakes.

Authors growth ecotypes form different origin under similar growth conditions and these ecotypes at day 21 have different developmental stages (number or rosette leaf, number and root length etc).

It is well known that hormone contents were highly variable between different cell type and organs in planta. It is known that different ecotype have different morphology at the same day from germination. For example, WS genotype can developed much faster as C24, have different number of rosette leaf before flowering. So, as material for hormone determination authors may used different “ratio between cell types”. Moreover, ecotype may have a different number of lateral root what definitely have a significant effect

From this point of view authors have to describe, at least briefly, status of material at the moment of harvest.

It is not so easy to follow figures to compare ecotypes. May be it is better to present it as table, but up to authors.

Table 1 described average temperature and precipitation, but the most important is these numbers for vegetation period. Plants do not growth in winter period for example.

Line 96: Ws-0 is a genotype with shorter vegetation period. How do hormone contents related with this feature? How rapid developed other ecotypes?  Please, mention these points in discussion.

Line 244: the response to stress regulated by hormonal contents is certain cell types, but in less manner in whole organs.

Line 282: please, clarify how much MES has been used.C

Author Response

Please find our point-by-point response in the attached file. Thank you.

Reviewer 2 Report

The manuscript deals with a relevant subject to PLANTS. The ms is very interesting and clear, well written, with an interesting set of well-presented results, and the appropriate topics are supported by the literature. I recommend that the manuscript should be accepted after minor revision.

Specific points:

Legend of Table 1: Mean annual precipitation instead of Mean monthly precipitation.

Table 1: Ecotype C24: Authors should remove Universidade de...(only Coimbra).

Table 1: Ecotype Cvi-0: Authors should be more precise with the region in Cape Verde Islands.

Line 101: Authors should use drought stress instead of stress.

The results and discussion subsection 2.1 should be improved in the discussion component.

Line 274: Authors should provide more information about supplemented light (type of lamps, light intensity, duration...).

Line 287: Authors should use photosynthetic photon flux density instead of photosynthetic radiation.

Line 443: random instead of radom.

Author Response

(The authors gave the same response as above.)

Round 2

Reviewer 1 Report

I think the autghors answered all points and manuscript can be accepter .

It will be greta if authors can calculate auxin/cytokinin ratio and provide link of speed of development (flowering time transition) and it correlation with hormnes contents/ratio.

Author Response

Thank you for an additional suggestion. For those ecotypes where data were available, we tested the correlations all hormonal contents and their rations with mean flowering time (as a marker of inherent ontogenetic development of ecotype). However, none of the tested relationships was significant. We included this conclusion briefly in the results section. We believe that this result adds to evidence that the differences in the rate of ontogenetic development among ecotypes had a negligible effect on results of our experiments.